# Entropy Estimators in SAR Image Classification

**DOI:** 10.3390/e24040509

**Published:** 2022-04-05

**Authors:** Julia Cassetti, Daiana Delgadino, Andrea Rey, Alejandro C. Frery

**Affiliations:** 1Instituto del Desarrollo Humano, Universidad Nacional de General Sarmiento, Los Polvorines B1613, Provincia de Buenos Aires, Argentina; 2Instituto de Ciencias, Universidad Nacional de General Sarmiento, Los Polvorines B1613, Provincia de Buenos Aires, Argentina; ddelgadino@campus.ungs.edu.ar; 3Centro de Procesamiento de Señales e Imágenes, Department of Mathematics, Universidad Tecnológica Nacional Facultad Regional Buenos Aires, Ciudad de Buenos Aires C1179AAQ, Argentina; arey@frba.utn.edu.ar; 4School of Mathematics and Statistics, Victoria University of Wellington, Wellington 6140, New Zealand; alejandro.frery@vuw.ac.nz

**Keywords:** feature extraction, synthetic aperture radar, Shannon entropy estimator, classification

## Abstract

Remotely sensed data are essential for understanding environmental dynamics, for their forecasting, and for early detection of disasters. Microwave remote sensing sensors complement the information provided by observations in the optical spectrum, with the advantage of being less sensitive to adverse atmospherical conditions and of carrying their own source of illumination. On the one hand, new generations and constellations of Synthetic Aperture Radar (SAR) sensors provide images with high spatial and temporal resolution and excellent coverage. On the other hand, SAR images suffer from speckle noise and need specific models and information extraction techniques. In this sense, the $G^{0}$ family of distributions is a suitable model for SAR intensity data because it describes well areas with different degrees of texture. Information theory has gained a place in signal and image processing for parameter estimation and feature extraction. Entropy stands out as one of the most expressive features in this realm. We evaluate the performance of several parametric and non-parametric Shannon entropy estimators as input for supervised and unsupervised classification algorithms. We also propose a methodology for fine-tuning non-parametric entropy estimators. Finally, we apply these techniques to actual data.

## 1. Introduction

Images obtained with coherent illumination systems, such as Synthetic Aperture Radar (SAR), are contaminated by speckle. This noise-like interference phenomenon corrupts the image in a non-Gaussian and non-additive manner, making difficult its processing and visual interpretation.

Against this backdrop, statistical procedures are essential tools for processing SAR data. A suitable model to describe this sort of image is fundamental to obtain features that promote a good analysis. In this sense, the family of $G^{0}$ distributions [1] has been extensively used to model SAR data because of its analytical simplicity and ability to describe a wide variety of roughness targets.

The application of machine and deep learning techniques to the problem of classification, segmentation and detection of objects in SAR images became more popular in recent times. Palacio et al. [2] used machine learning techniques in combination with filters to perform classification in PolSAR images. Baek and Jung [3] carried out a comparison between three different machine learning techniques to classify single- and dual-pol SAR image showing that the deep neural network presented the best performance.

Different authors used methods based on transfer learning techniques to classify SAR images. These methods aim to solve the problem of having limited labeled area information to train deep convolutional neural networks (CNN). Kang and He [4] applied this technique using a CNN trained on a CIFAR-10 dataset to extract a mid-level representation. They showed that this technique is adequate to solve the problem of the limited amount of labeled SAR data, by comparing the results obtained with a CNN without using this technique and combining a Support Vector Machine (SVM) with a Gabor filter or with gray level co-occurrence matrices. Lu and Li [5] implemented this methodology using several popular pre-trained models and proposed a new method of data augmentation. They also made a comparison with some related works and showed that their proposed method outperformed the others. Huang et al. [6] proposed to transfer the knowledge obtained from a large number of unlabeled SAR images by incorporating a reconstruction path with stacked convolutional autoencoders in the network architecture. Their proposal was competitive for the MTSAR dataset using all training samples, and had the best performance when the training dataset has a small size.

Transfer learning was also implemented by Rostami et al. [7]. They proposed to transfer the knowledge from the electro-optical domain to SAR by learning a shared embedding space, and they showed that their approach is effective when applied to a ship classification problem. Huang et al. [8] proposed another deep transfer learning method to solve the land cover classification problem with highly unbalanced classes, geographic diversity and noisy labels. They showed that the proposed model, which uses cross-entropy, can be generalized and can be applied to others SAR domains.

Several approaches have been developed in order to obtain expressive and tractable features from SAR data. In particular, entropy measures have been widely used for this purpose. Parameter estimation [9], classification [10], procedures for constructing confidence interval and contrast measures [11,12], edge detection [13], and noise reduction filters [14] are among their applications.

Sundry authors have tackled the segmentation and classification SAR images problem using information theory measures. Nobre et al. [15] used Rényi’s entropy for monopolarized SAR image segmentation. Ferreira and Nascimento [16] derived a closed-form expression for the Shannon entropy based on the $G^{0}$ law for intensity data and proposed a new entropy-based segmentation method. Carvalho et al. [10] employed stochastic distances to approach unsupervised classification applied to Polarimetric Synthetic Aperture Radar (PolSAR) images. Shannon entropy has been applied to analyzed SAR imagery in several approaches, from inference [11] to classification [16]. Therefore, its estimation deserves attention.

The parametric expression of the Shannon entropy for a system characterized by a continuous random variable is the following well-known expression:(1)$$H\left(Z\right)=-E\left[\log f\left(z\right)\right]=-\int_{R}f\left(z\right)\log f\left(z\right)dz$$
where *f* is the probability density function that characterizes the distribution of the real-valued random variable *Z*. Several procedures can be applied to obtain an estimate of H(Z) given a random sample $Z=(Z_{1},Z_{2},\ldots ,Z_{n})$.

The most direct family of estimators of H(Z) given Z consists of obtaining estimators for θ, the parameter that indexes the distribution of *Z*, say $\hat{\theta }$, and using them in (Equation 1). This approach yields the families of maximum likelihood, moments, and robust estimators, to name a few. This is the “parametric approach”.

“Non-parametric” approaches do not use $\hat{\theta }$ as a proxy. Instead, they rely on the equivalent expression for the Shannon entropy given by
(2)$$H\left(Z\right)=\int_{0}^{1}\log\frac{dF^{-1}\left(p\right)}{dp}dp,$$
where *F* is the cumulative distribution function that also characterizes the distribution of the random variable [17]. Such alternative approaches compute estimates of *F* in Equation (Equation 2) from the observed sample. Vasicek [17] replaced the distribution function *F* by the empirical distribution function $F_{n}$ and used a difference operator in place of the differential operator. van Es [18] studied an entropy estimator based on differences between order statistics. Correa [19] proposed a new entropy estimator determined from local linear regression. Al-Omari [20] and Noughabi and Noughabi [21] presented modified versions of the estimator introduced by Ebrahimi et al. [22].

It is important to mention that these estimators have been studied in different contexts. Maurizi [23] studied the works by Vasicek [17] and van Es [18] to estimate the entropy H(Z) when the random variable has support [0,1]. Noughabi and Park [24] considered them to propose goodness of fit tests for the Laplace distribution. Suk-Bok et al. [25] assessed the proposal by [17] to estimate H(Z) for a double exponential function in the framework of multiple type-II censored sampling. More recently, Al-Labadi et al. [26] considered these estimators to propose a new Bayesian non-parametric estimation to entropy. Additionally, Lopes and Machado [27] considered Ref. [22] as a reference in the review of other entropy estimators.

In this paper, we study the performance of parametric and non-parametric estimators of the entropy in the context of supervised and unsupervised classification. In the parametric case, we use the relationship between the $G^{0}$ and Fisher distributions to obtain an expression of the entropy. In the non-parametric case, we assess these estimators in terms of bias, mean square error, computational time, and accuracy.

## 2. Materials and Methods

### 2.1. The $G^{0}$ Model

The multiplicative model defines the return *Z* in a monopolarized SAR image as the product of two independent random variables: one corresponding to the backscatter *X*, and the other to the speckle noise *Y*. In this manner, Z=XY represents the return in each pixel of the image.

The $G^{0}$ distribution is an attractive model for *Z* because of its flexibility to adequately model areas with all types of roughness [28,29]. For intensity SAR data, this family arises from considering the speckle noise *Y* modeled as a Γ distributed random variable with unitary mean and the shape parameter L≥1, the number of looks. We also assume that the backscatter *X* obeys a reciprocal gamma law. Thus, the density function for intensity data is given by
$$f\left(z\right)=\frac{L^{L}\Gamma \left(L-\alpha \right)}{\gamma^{\alpha }\Gamma \left(-\alpha \right)\Gamma \left(L\right)}\;\cdot \;\frac{z^{L-1}}{{\left(\gamma +zL\right)}^{L-\alpha }},$$
where −α,γ,z>0 and L≥1. The *r*-order moment is
(3)$$E\left(Z^{r}\right)={\left(\frac{\gamma }{L}\right)}^{r}\frac{\Gamma (-\alpha -r)}{\Gamma (-\alpha )}\;\cdot \;\frac{\Gamma (L+r)}{\Gamma (L)},$$
provided α<−r, and infinite otherwise.

Mejail et al. [28] proved a relationship between the $G^{0}$ distribution and the Fisher–Snedekor *F* law, which states that the cumulative distribution function $F_{\alpha ,\gamma ,L}$ for the return *Z* is
(4)$$F_{\alpha ,\gamma ,L}\left(z\right)=\Upsilon_{2L,-2\alpha }\left(-\alpha z/\gamma \right),$$
for every z>0, where $\Upsilon_{2L,-2\alpha }$ is the cumulative distribution function of a Fisher–Snedekor random variable with 2L and −2α degrees of freedom. This connection is helpful to obtain a closed formula for the entropy.

### 2.2. Shannon Entropy

Shannon’s contribution to the creation of what is known as information theory is well known. Shannon [30] proposed a new way of measuring the transmission of information through a channel, thinking of information as a statistical concept. The entropy of the $G^{0}$ distribution can be obtained using (Equation 4). Denote $H_{F}$ as the entropy under the Fisher–Snedekor model; then the $G^{0}$ entropy for intensity data $H_{G^{0}}$ is
(5)$$H_{G^{0}}\left(\alpha ,\gamma ,L\right)=H_{F}\left(2L,-2\alpha \right)-\log\left(-\alpha /\gamma \right).$$
Using (Equation 5), the expression of $H_{G^{0}}$ is
(6)$$\begin{array}{c} H_{G^{0}}\left(\alpha ,\gamma ,L\right)=-\log\left(-\alpha /\gamma \right)-\left(1-\alpha \right)\psi^{\left(0\right)}\left(-\alpha \right)+\log\left(-\alpha /L\right)+\left(L-\alpha \right)\psi^{\left(0\right)}\left(L-\alpha \right)\\ +\log\left(B(L,-\alpha )\right)+\left(1-L\right)\psi^{\left(0\right)}\left(L\right), \end{array}$$
where $\psi^{\left(0\right)}$ and *B* are the digamma and beta functions, respectively.

Figure 1 shows the theoretical entropy $H_{G^{0}}\left(\alpha ,\gamma ,L\right)$ as a function of α and γ with L=2. It can be shown that for each fixed γ value, $H_{G^{0}}$ is an injective function. The same behavior repeats if we consider α as a constant.

### 2.3. Shannon Entropy Estimators

Several authors have proposed entropy estimators using (Equation 2). Most of them are based on order statistics of the sample. Al-Omari [20] presented an overview of these estimators and also proposed a new one. From a parametric point of view, it is natural to consider the maximum likelihood estimator (ML) of the entropy ($H_{\mathrm{ML}}$).

In what follows, we describe the entropy estimators studied in this paper.

#### 2.3.1. Maximum Likelihood Entropy Estimator

Let $Z=(Z_{1},\ldots ,Z_{n})$ be an independent random sample of size *n* from the $G^{0}\left(\alpha ,\gamma ,L\right)$ distribution. Assume that *L* is known. The maximum likelihood estimator of (α,γ) for *L* is known and denoted as $\left({\hat{\alpha }}_{\mathrm{ML}},{\hat{\gamma }}_{\mathrm{ML}}\right)$, which consists of the values in the parametric space $R_{-}\times R_{+}$, which maximize the reduced log-likelihood function: (7)$$\ell \left(\alpha ,\gamma ;L,Z\right)=\log\Gamma \left(L-\alpha \right)-\alpha \log\gamma -\log\Gamma \left(-\alpha \right)+\frac{\alpha -L}{n}\sum_{i=1}^{n}\log\left(\gamma +LZ_{i}\right).$$
Solving (Equation 7) requires numerical maximization routines that, under certain circumstances, do not converge [31]. We use the L-BFGS-B version of the Broyden–Fletcher–Goldfarb–Shannon (BFGS) method [32] that allows box constraints. This algorithm belongs to the quasi-Newton methods family, not requiring the Hessian matrix but only the gradient. The optimal asymptotic properties of the ML estimator are well-known.

The ML entropy estimator [33] is
(8)$$\begin{array}{c} {\hat{H}}_{\mathrm{ML}}\left(Z\right)=H_{G^{0}}\left({\hat{\alpha }}_{\mathrm{ML}},{\hat{\gamma }}_{\mathrm{ML}},L\right). \end{array}$$
This estimator inherits all of the good properties of ML estimators (consistency and asymptotic normality), but also their pitfalls: sensitivity to the initial value, lack of convergence due to flatness of (Equation 7), and lack of robustness. Convergence problems, which are more prevalent with small samples and with data from textureless areas, were identified by Frery et al. [31] and mitigated with a line-search algorithm. Refs. [9,34,35] studied robust alternatives to (Equation 7).

#### 2.3.2. Non-Parametric Entropy Estimators

Assume that $Z=(Z_{1},Z_{2},\ldots ,Z_{n})$ is a random sample from the law characterized by the distribution function F(z) whose order statistics are $Z_{\left(1\right)},Z_{\left(2\right)},\ldots ,Z_{\left(n\right)}$. Vasicek [17] proposed the following entropy estimator: (9)$${\hat{H}}_{V}\left(Z\right)=\frac{1}{n}\sum_{i=1}^{n}\log\left[\frac{n}{2m}\left(Z_{\left(i+m\right)}-Z_{\left(i-m\right)}\right)\right],$$
with m<n/2 as a positive integer, $Z_{\left(i+m\right)}-Z_{\left(i-m\right)}$ the spacing of order *m*, or *m*-spacing, $Z_{\left(i\right)}=Z_{\left(i\right)}$ if 1<i, and $Z_{\left(i\right)}=Z_{\left(n\right)}$ if i>n. The author proved that this estimator is weakly consistent for H(Z) when m/n→0 and n,m→∞.

The only possible numerical problem with this estimator and its variants is having zero as the argument of the logarithm, a situation that can be easily checked and solved. Their computational complexity reduces to adding differences of order statistics. These estimators are robust by nature, since they do not depend on any particular model. Differently from the approaches discussed in Refs. [9,34,35], achieving such a robustness does not impose a heavy computational burden.

Several authors introduced modifications to Vasicek’s estimator. In this work we consider the following entropy estimators variants, surveyed by Al-Omari [20].

van Es [18]:
(10)$$\begin{array}{c} {\hat{H}}_{\mathrm{VE}}\left(Z\right)=\frac{1}{n-m}\sum_{i=1}^{n-m}\log\left[\frac{n+1}{m}\left(Z_{\left(i+m\right)}-Z_{\left(i\right)}\right)\right]+\sum_{k=m}^{n}\frac{1}{k}+\log\frac{m}{n+1}. \end{array}$$Correa [19]:
(11)$${\hat{H}}_{C}\left(Z\right)=-\frac{1}{n}\sum_{i=1}^{n}\log\frac{\sum_{j=i-m}^{i+m}\left(j-i\right)\left(Z_{\left(j\right)}-{\bar{Z}}_{\left(i\right)}\right)}{n\sum_{j=i-m}^{i+m}{\left(Z_{\left(j\right)}-{\bar{Z}}_{\left(i\right)}\right)}^{2}},$$
where ${\bar{Z}}_{\left(i\right)}={\left(2m+1\right)}^{-1}\sum_{j=i-m}^{i+m}Z_{\left(j\right)}$.Noughabi and Arghami [36]:
(12)$${\hat{H}}_{\mathrm{NA}}\left(Z\right)=\frac{1}{n}\sum_{i=1}^{n}\log\left[\frac{n}{c_{i}m}\left(Z_{\left(i+m\right)}-Z_{\left(i-m\right)}\right)\right]$$
where
$$c_{i}=\left\{\begin{array}{cc} 1 & \;\mathrm{if}\;1\leq i\leq m,\\ 2 & \;\mathrm{if}\;m+1\leq i\leq n-m,\\ 1 & \;\mathrm{if}\;n-m+1\leq i\leq n, \end{array}\right.$$
and $Z_{\left(i-m\right)}=Z_{\left(1\right)}$ if i≤m and $Z_{\left(i+m\right)}=Z_{\left(n\right)}$ for i≥n−m.Al-Omari [37]:
(13)$${\hat{H}}_{{\mathrm{AO}}_{1}}\left(Z\right)=\frac{1}{n}\sum_{i=1}^{n}\log\left[\frac{n}{\omega_{i}m}\left(Z_{\left(i+m\right)}-Z_{\left(i-m\right)}\right)\right],$$
where
$$\omega_{i}=\left\{\begin{array}{cc} 3/2 & \;\mathrm{if}\;1\leq i\leq m,\\ 2 & \;\mathrm{if}\;m+1\leq i\leq n-m,\\ 3/2 & \;\mathrm{if}\;n-m+1\leq i\leq n, \end{array}\right.$$
in which $Z_{\left(i-m\right)}=Z_{\left(1\right)}$ for i≤m, and $Z_{\left(i+m\right)}=Z_{\left(n\right)}$ for i≥n−m.Al-Omari alternative proposal [20]:
(14)$${\hat{H}}_{{\mathrm{AO}}_{2}}\left(Z\right)=\frac{1}{n}\sum_{i=1}^{n}\log\left[\frac{n}{v_{i}m}\left(Z_{\left(i+m\right)}-Z_{\left(i-m\right)}\right)\right],$$
where
$$v_{i}=\left\{\begin{array}{cc} 1+(i-1)/m & \;\mathrm{if}\;1\leq i\leq m,\\ 2 & \;\mathrm{if}\;m+1\leq i\leq n-m,\\ 1+(n-i)/2m & \;\mathrm{if}\;n-m+1\leq i\leq n, \end{array}\right.$$
in which $Z_{\left(i-m\right)}=Z_{\left(1\right)}$ for i≤m, and $Z_{\left(i+m\right)}=Z_{\left(n\right)}$ for i≥n−m.Ebrahimi et al. [22]:
(15)$$\begin{array}{c} {\hat{H}}_{E}\left(Z\right)=\frac{1}{n}\sum_{i=1}^{n}\log\left[\frac{n}{\tau_{i}m}\left(Z_{\left(i+m\right)}-Z_{\left(i-m\right)}\right)\right], \end{array}$$
where
$$\tau_{i}=\left\{\begin{array}{cc} 1+(i-1)/m & \;\mathrm{if}\;1\leq i\leq m,\\ 2 & \;\mathrm{if}\;m+1\leq i\leq n-m,\\ 1+(n-i)/m & \;\mathrm{if}\;n-m+1\leq i\leq n. \end{array}\right.$$

van Es [18] showed that, under general conditions, (Equation 10) converges almost surely to H[Z] when m,n→∞, m/log(n)→∞, and m/n→0. The author also proved the estimator’s asymptotic normality when m,n→∞ and $m=o(n^{1/2})$. Correa [19], through a simulation study, showed that his estimator has a smaller mean squared error than Vasiciek’s proposal (Equation 9).

Al-Omari’s estimators, cf. (Equation 13) and (Equation 14), converge in probability to H[Z] when m,n→∞andm/n→0. Ebrahimi et al. [22] presented an estimator adjusting Vasicek’s [17] weight. Under the same conditions as Al-Omari [37], the authors proved that ${\hat{H}}_{E}\left(Z\right)\underset{n\to \infty }{\overset{p}{⟶}}H\left[Z\right]$ when m,n→∞andm/n→0. The same applies to the Noughabi–Arghami estimator.

### 2.4. Estimator Tuning

The choice of the spacing parameter *m* in this type of estimators is an important task that is still open. Wieczorkowski and Grzegorzewski [38] proposed the following heuristic formula:(16)$$m_{\mathrm{WG}}=\left[\sqrt{n}+0.5\right].$$

Our goal is to find a value of *m* that performs well in a range of parameters α and sample sizes *n* when estimating the entropy under the $G^{0}$ model. In order to achieve this goal, we assess the performance of (Equation 16) with a Monte Carlo study for each one of the entropy estimators presented in Section 2.3.2 under the $G^{0}$ model. We considered a parameter space comprised of:Sample sizes $n\in \left\{9,25,49,81,121\right\}$, which represent different scenarios of squared windows of sides 3, 5, 7, 9 and 11;Texture values $\alpha \in \left\{-8,-5,-3,-1.5\right\}$ to depict areas with different levels of roughness and L=2 (the L=1 case was studied by Cassetti et al. [39]).

Since γ is a scale parameter, we based the forthcoming analysis on the condition E(Z)=1, which links texture and brightness by $\gamma^{*}=-\alpha -1$. With the aim to simplify the notation, we consider $\alpha_{j}$ with j=1,2,3,4 where $\alpha_{1}=-1.5$, $\alpha_{2}=-3$, $\alpha_{3}=-5$ and $\alpha_{4}=-8$. Thus, $\gamma_{j}^{*}=-\alpha_{j}-1$.

For each fixed *n* and *j* we draw 1000 independent samples $z_{1},\ldots ,z_{n}$ from $G^{0}\left(\alpha_{j},\gamma_{j}^{*},2\right)$. We used $m=m_{\mathrm{WG}}$ and calculated all estimators ${\hat{H}}_{mj}^{i}$ from Section 2.3.2. Therefore, we obtained a vector of estimates $\left({\hat{H}}_{mj}^{1},{\hat{H}}_{mj}^{2},\ldots ,{\hat{H}}_{mj}^{1000}\right)$ from which we computed the sample mean ${\bar{\hat{H}}}_{mj}=1000^{-1}\sum_{i=1}^{1000}{\hat{H}}_{mj}^{i}$, the sample bias ${\hat{B}}_{mj}={\bar{\hat{H}}}_{mj}-H_{j}$, where $H_{j}$ is the true entropy from (Equation 6), and the sample mean squared error ${\hat{\mathrm{MSE}}}_{mj}=1000^{-1}\sum_{i=1}^{1000}{\left({\hat{H}}_{mj}^{i}-H_{j}\right)}^{2}$. Then, we analyzed the performance of these estimators in terms of bias and MSE.

In order to improve the spacing (Equation 16), we implemented another strategy to choose, for each sample size *n*, the best value *m* to be used for all textures α. In the following, we considered m∈{1,2,…,⌊n/2⌋} as was indicated in (Equation 9). We repeated the same methodology as before for each *m* and for each *j*, obtaining $\left\{{\hat{B}}_{1j},\ldots ,{\hat{B}}_{\lfloor n/2\rfloor j}\right\}$. This vector is represented in the *j*th column of Table 1. We then calculated, in each row of the table, the average of the absolute value of bias (shown in the last column of Table 1). The best *m* value is $m=\underset{s}{arg min}\bar{|{\hat{B}}_{s.}|}$. Table 1 shows the schema of the methodology employed, for fixed *n* and an entropy estimator. Each table entry, ${\hat{B}}_{sj}$, represents the bias for m=s and $\alpha =\alpha_{j}$.

Section 3.1 presents the results of this approach. The spacing values we obtained are different from the heuristic formula (Equation 16), and they lead to better estimates in terms of bias and mean squared error.

### 2.5. Classification

To study the performance of the selected entropy estimators in terms of SAR image classification, we divided the analysis into simulated and actual images. We used unsupervised and supervised techniques to choose the three estimators that led to the best values of classification quality. For the former, we applied a *k*-means algorithm, which groups data into *k* classes setting *k* centroids and minimizing the variance within each group. This non-hierarchical clustering technique has been applied in many studies in SAR image processing, cf. the works by Niharika et al. [40] and by Liu et al. [41].

For the latter approach we implemented a support vector machine (SVM) algorithm, which is a supervised machine learning technique [42] whose objective is to define, given a set of features, the best possible separation between classes by finding a hyperplane that maximizes the margin of separation between these classes. It is common to accept some misclassification to obtain a better overall performance; this is achieved through the penalizing parameter *c*.

When data cannot be separated by a hyperplane, they are transformed to a higher-dimensional feature space through a suitable non-linear transformation called “kernel function”. Given $x,x^{\prime }\in R^{n}$, linear and radial kernels are respectively defined by $K_{L}\left(x,x^{\prime }\right)=\langle x,x^{\prime }\rangle$ and $K_{R}\left(x,x^{\prime }\right)=\exp\left(-g{∥x-x^{\prime }∥}^{2}\right)$, for g>0.

We randomly selected 1000 pixels in each of the four regions, far away enough from the boundaries, to find the best kernel and hyperparameters. This reference sample was divided into two sets: training and validation (80% of the sample), and testing (20%). We considered linear and radial types for the kernel, with the penalizing parameters c=0.001,0.01,0.1,1,5,10 and g=0.01,0.1,1,1.5,2. With the training–validation set we made a 5-cross fold validation, and computed the mean and the standard deviation of the F1-scores. Recall that F1=2·TPR·PPV/(TPR+PPV), where TPR is the True Positive Rate and PPV is the Positive Predictive Value.

This approach has been applied in different areas, such as sea oil spill monitoring [43], pattern recognition [44], and classification of polarimetric SAR data [2], among other applications.

We used different measures of quality depending on the type of classification. In the unsupervised case, we used the Calinski–Harabasz (CH) [45] and Davies–Bouldin [46] (DB) indexes, while we present the Kappa coefficient for the supervised classification. We also show the accuracy of both algorithms. All of these measures should be interpreted as “bigger is better”, except for the DB index, for which “lower is better”.

## 3. Results and Discussion

### 3.1. Choice of the Spacing Parameter m for Non-Parametric Estimators

Figure 2 presents the bias and the MSE for the Wieczorkowski and Grzegorzewski [38] criterion, L=2 case, and for all of the estimators analyzed, except for the Al-Omari (Equation 14) and Ebrahimi (Equation 15) estimators. These two estimators presented large bias and, thus, were discarded for further analysis.

It can be seen that there is no single estimator that performs best for all α values, but $H_{C}$ and $H_{{\mathrm{AO}}_{1}}$ present low bias and low MSE for all of the cases studied except for α=−1.5. The others estimators show bad behavior in terms of bias because of their slower convergence to zero for all of the cases studied.

Table 2 shows the best *m* chosen according to the methodology used for L=1 and L=2, for samples coming from $G_{0}$ distribution.

Notice that, with few exceptions, the optimal spacing *m* is smaller than the empirical formula $m_{\mathrm{WG}}$.

### 3.2. Performance of the Nonparametric Estimators for the Selected m Value

In order to study the behavior of our proposal for the selection of the *m* value we performed a Monte Carlo simulation as described in Section 2.4. Figure 3 shows the results obtained for the estimators studied for the *m* value chosen in terms of bias and MSE, for L=2 case. We also plotted the $H_{\mathrm{ML}}$ estimator. It can be observed that there is an improvement in entropy estimation in terms of bias and MSE with our methodology, compared to the (Equation 16) heuristic formula for all of the estimators studied. All of them show a faster convergence of the bias to zero and are competitive with the performance of the $H_{\mathrm{ML}}$ estimator in terms of bias and MSE, for sample sizes larger than 81.

As mentioned, the optimized spacing leads, in most cases, to the use of more samples than the (Equation 16) criterion. This suggests that the latter is an optimistic view of the information content of each sample, at least when dealing with $G^{0}$ deviates. In other words, theses observations are less informative for the estimation of the entropy. Because of this, a smaller spacing, i.e., larger samples, are required to achieve good estimation quality.

In the following, we present empirical results classifying a simulated image SAR.

### 3.3. Simulated Image

We generated two 300×300 images with observations coming from $G^{0}$ distributions with L=1,2, γ=0.1, and four different classes: α∈{−1.5,−3,−5,−8}. Figure 4a shows the image obtained with L=2, where the brightest area corresponds to α=−1.5, i.e., extremely textured observations. As the brightness decreases, the texture changes from heterogeneous (α=−3 and −5) to a homogeneous zone corresponding to the darkest area (α=−8). As the performance measures were similar in both L=1,2 cases, i.e., single and multi-look, we only show results from the latter.

We computed a map of estimated entropies ($\hat{H}$) with each estimator by sweeping the image with sliding windows of sizes s×s, for s=3,5,7,9,11. These are the sample sizes studied in Section 2.4. Then, we used $\hat{H}$ as a feature to classify by both the unsupervised and supervised techniques.

Figure 4b shows the result of classifying by the *k*-means algorithm the ${\hat{H}}_{C}$ map of values obtained with s=9.

Figure 4c shows the accuracy as a function of the sample size. It can be observed that a 9×9 window presents the best accuracy. It can also be seen that the ${\hat{H}}_{\mathrm{ML}}$ estimator has the worst performance, whereas ${\hat{H}}_{C}$, ${\hat{H}}_{\mathrm{NA}}$ and ${\hat{H}}_{\mathrm{VE}}$ show the best performance. These results are corroborated by the values shown in Table 3, in which the best performances are shown in bold font.

Table 4 presents the CH and DB values for the best sample size (s=9, n=81). According to CH, ${\hat{H}}_{C}$, ${\hat{H}}_{\mathrm{NA}}$, and ${\hat{H}}_{\mathrm{VE}}$ have the best performance, whereas DB selected ${\hat{H}}_{C}$, ${\hat{H}}_{\mathrm{NA}}$, and ${\hat{H}}_{V}$ as the best.

We also provide, for the sake of comparison, quality measures obtained by $H_{\mathrm{ML}}$.

Table 5 shows the selected kernels and hyper-parameters that maximize the F1 mean and minimize the F1 variance. The best models were trained using the whole reference sample and applied to classify the complete image. The accuracy and κ coefficient were computed, and the results are shown in Figure 5. The best accuracy values are shown in Table 6, as well as the models that achieved them. It can be seen that the optimal value for L=1 was obtained for a sliding window of size 9×9. Sizes 9×9 and 7×7 presented similar (best) values. In this sense, with the purpose of providing a unified criterion, we chose the size of the sliding window as 9×9 to perform the analysis.

Table 7 and Table 8 show the confusion matrices when the models are applied to the simulated images, L=1,2. It can be observed that if L=1, ${\hat{H}}_{C}$, ${\hat{H}}_{\mathrm{NA}}$, and ${\hat{H}}_{V}$ overcame ${\hat{H}}_{\mathrm{ML}}$ for extremely high, high, and middle textured areas, respectively. For L=2, ${\hat{H}}_{\mathrm{ML}}$ performed better than the other models except for regions with a very high level of texture in which ${\hat{H}}_{V}$ and ${\hat{H}}_{{\mathrm{AO}}_{1}}$ produced better results.

### 3.4. Actual Images

We assessed our proposal with two SAR images. First, we considered an image of the surroundings of Munich in Germany of the size 459×494, which was acquired in L-band, HV polarization, and complex single look format. Second, we used a subsample of 500×645 pixels of a full PolSAR image of California’s San Francisco bay area, taken by the NASA/JPL AIRSAR L-band instrument in intensity format.

We applied the SVM algorithm to both actual images, replicating the procedure described in the study of simulated data, using the entropy estimator as a feature for classification of the three polarizations.

The Equivalent Number of Looks (ENL) using uncorrelated data is defined as $\mathrm{ENL}=1/\hat{{\mathrm{CV}}^{2}}$, the reciprocal of the sample coefficient of variation $\hat{\mathrm{CV}}=\hat{\sigma }/\hat{\mu }$, where $\hat{\sigma }$ is the sample standard deviation and $\hat{\mu }$ is the sample mean [47]. In order to find the ENL in each polarization band of the image of San Francisco, we manually selected samples from homogeneous areas in each band and calculated ENL as an average weighted by the sample size per band. Finally, the ENL is the average of the estimations in each polarization. We obtained 2.53, 3.41, and 3.41 as the ENL values in the HH, HV, and VV bands, respectively. Thus, we considered the ENL as equal to 3.12 for the whole image. We then used the same spacings, *m*, for L=2 and L=3.

Figure 6 and Figure 7 show the training samples selected to perform the supervised classification in both images. In the fist case, we worked with three types of regions: urban (red), forest (dark green), and pasture (light green). In the other case, we selected five areas: water (blue), urban zone (red), vegetation (green), pasture (yellow), and beach (orange).

We studied linear and radial kernels; the last one produced better results, except for ${\hat{H}}_{{\mathrm{AO}}_{1}}$ and ${\hat{H}}_{V}$ when applied to the image of Munich. The combinations of hyper-parameters are the following:c=1 for ${\hat{H}}_{{\mathrm{AO}}_{1}}$ in Munich;c=1 and g=0.1 for ${\hat{H}}_{C}$ in Munich;c=0.01 and g=1 for ${\hat{H}}_{\mathrm{NA}}$ in Munich;c=10 for ${\hat{H}}_{V}$ in Munich;c=5 and g=1.5 for ${\hat{H}}_{\mathrm{VE}}$ in Munich;c=5 and g=1.5 for ${\hat{H}}_{\mathrm{ML}}$ in Munich;c=1 and g=0.1 for ${\hat{H}}_{{\mathrm{AO}}_{1}}$ in San Francisco;c=10 and g=1.5 for ${\hat{H}}_{C}$ in San Francisco;c=5 and g=2 for ${\hat{H}}_{\mathrm{NA}}$ in San Francisco;c=5 and g=2 for ${\hat{H}}_{V}$ in San Francisco;c=10 and g=2 for ${\hat{H}}_{\mathrm{VE}}$ in San Francisco;c=10 and g=1.5 for ${\hat{H}}_{\mathrm{ML}}$ in San Francisco.

We subsequently included the CV as a feature in the classification process. In this case, the best performance was achieved for the linear kernel with a cost of 10 applied to the image of San Francisco, except for ${\hat{H}}_{{\mathrm{AO}}_{1}}$ and ${\hat{H}}_{V}$ showing a best performance if a radial kernel is used with c=5 and g=1, respectively, and ${\hat{H}}_{\mathrm{ML}}$ with a radial kernel using c=10 and g=1.5, respectively. On the other hand, the radial kernel produced the best results for the image of Munich using the following hyper-parameters:c=1 and g=0.1 for ${\hat{H}}_{{\mathrm{AO}}_{1}}$;c=5 and g=2 for ${\hat{H}}_{C}$;c=1 and g=2 for ${\hat{H}}_{\mathrm{NA}}$;c=1 and g=0.1 for ${\hat{H}}_{V}$;c=10 and g=1 for ${\hat{H}}_{\mathrm{VE}}$;c=10 and g=1.5 for ${\hat{H}}_{\mathrm{ML}}$.

Table 9 and Table 10 present the test accuracy and Kappa index. We also show the validation accuracy, which was computed using cross-validation with five folds; these values are similar to the test accuracy, showing that there is no evidence of overfitting. In addition, we show that including the CV coefficient as a feature in the classification problem improved the results.

If we only consider the entropy, ${\hat{H}}_{\mathrm{VE}}$ showed the best performance in both single and multilook cases. However, if we add CV as a characteristic, then ${\hat{H}}_{C}$ appears as the best classifier followed by ${\hat{H}}_{\mathrm{NA}}$ and ${\hat{H}}_{\mathrm{ML}}$ for the single-look case, and ${\hat{H}}_{{\mathrm{AO}}_{1}}$ followed by ${\hat{H}}_{\mathrm{NA}}$ and ${\hat{H}}_{V}$ for the multilook case.

Figure 8 and Figure 9 exhibit the classification of the whole images when our proposal is applied. It can be observed that in the case of the image of San Francisco the classifiers distinguished the beach and, with the addition of the CV, some roads surrounded by trees were better classified.

The processing time is an important feature when proposing a new estimator. Table 11 shows the processing time, measured in minutes, needed to perform a map of estimated entropies moving through the image with sliding windows of size 9×9 for each one of the estimators applied to the Munich and San Francisco images. It can be seen that $H_{V}$ had the shortest processing time, followed by $H_{\mathrm{VE}}$ and $H_{\mathrm{NA}}$.

We conclude this section by comparing the results of classifying by using estimates of the entropy with those obtained with a classical approach. Table 12 compares the results obtained using our best models against the technique that applies the improved Lee filter [48] and then classifies using SVM. Figure 10 shows the classification of the whole images applying the alternative method. It can be observed that our proposal offers advantages that prior methods cannot.

## 4. Conclusions

We assessed the performance of six non-parametric entropy estimators in conjunction with the ML estimator in terms of bias, MSE and image classification for single and multilook cases.

On the one hand, the advantage of using these non-parametric estimators is that they are very simple to implement, since they do not assume any model and do not need optimization algorithms. On the other hand, they depend on a space parameter *m*. Although the literature recommends a heuristic value, we proposed a criterion for choosing the value of *m* that presents the slightest bias in the entropy estimation for all of the textured values studied and all of the sample sizes analyzed. This criterion presents better performance than that proposed by Wieczorkowski and Grzegorzewski [38].

With these values for *m*, we applied unsupervised (*k*-means) and supervised (SVM) classification algorithms to both simulated and actual data, and compared their performance with the ${\hat{H}}_{\mathrm{ML}}$ entropy estimator. We showed evidence that ${\hat{H}}_{\mathrm{VE}}$ presents the best performance in terms of accuracy and kappa index for both single and multilook cases, when it is applied to actual images. However, when we added the coefficient of variation as a feature used by the classifier, both measures improved and the best estimators changed. ${\hat{H}}_{C}$ and ${\hat{H}}_{{\mathrm{AO}}_{1}}$ performed the best for the single and multilook cases, respectively, showing an improvement of 1% for the former and of 3% for the latter. However, these two estimators require longer processing times than the others.

We completed the analysis by comparing our proposal with another technique that combines the improved Lee filter with an SVM classifier, showing that the entropy-based approach presents better accuracy indexes.

Hence, we strongly recommend to consider these non-parametric estimators because of the simplicity of their implementation and their good performance.

## Figures and Tables

**Figure 1 entropy-24-00509-f001:**
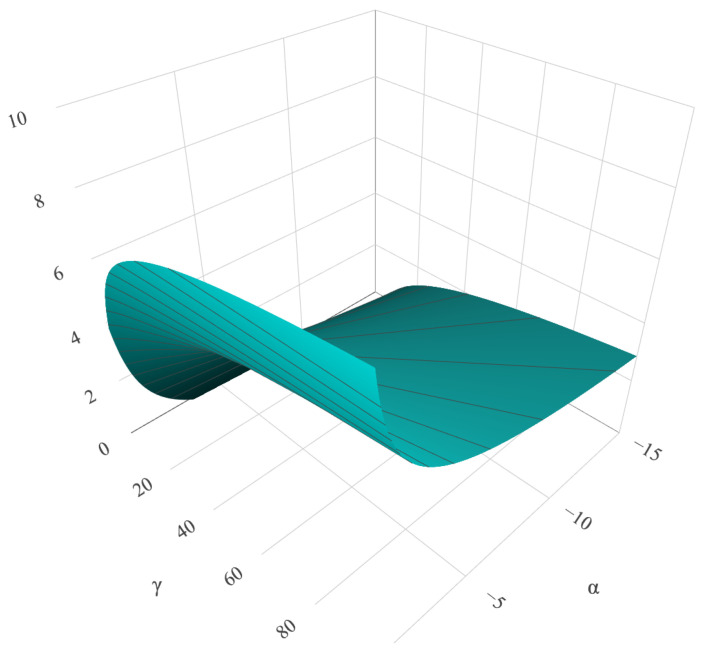
$H_{G^{0}}\left(\alpha ,\gamma ,L\right)$ as a function of α and γ for L=2.

**Figure 2 entropy-24-00509-f002:**
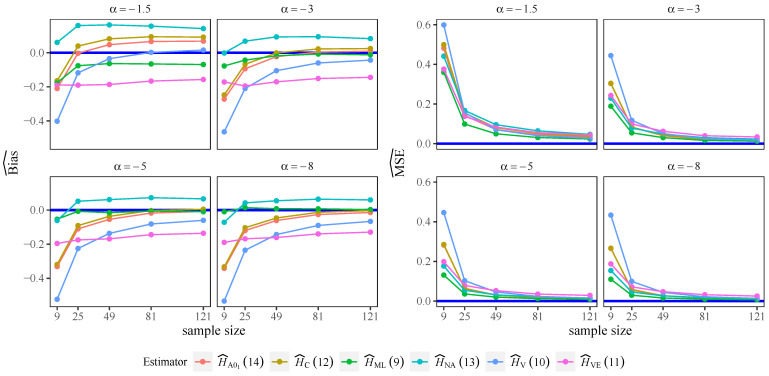
Bias and MSE for Wieczorkowski and Grzegorzewski [38] criterion given by (Equation 16), L=2.

**Figure 3 entropy-24-00509-f003:**
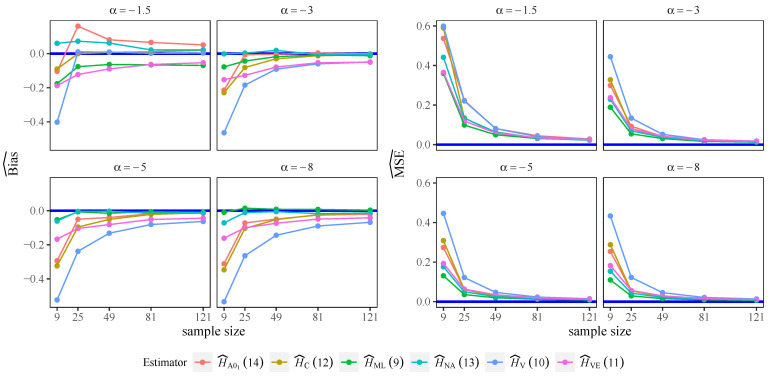
Bias and MSE for authors’ proposed *m* choice, L=2.

**Figure 4 entropy-24-00509-f004:**
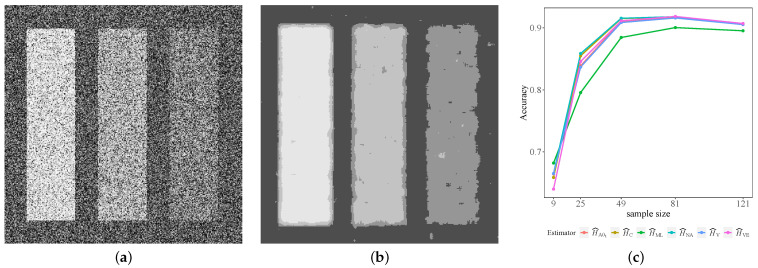
*K*-means applied to a simulated image with L=2, γ=0.1 and sliding windows size 9×9. (**a**) Simulated image. (**b**) Classification with ${\hat{H}}_{C}$ and s=9. (**c**) Accuracy as a function of the sample size.

**Figure 5 entropy-24-00509-f005:**
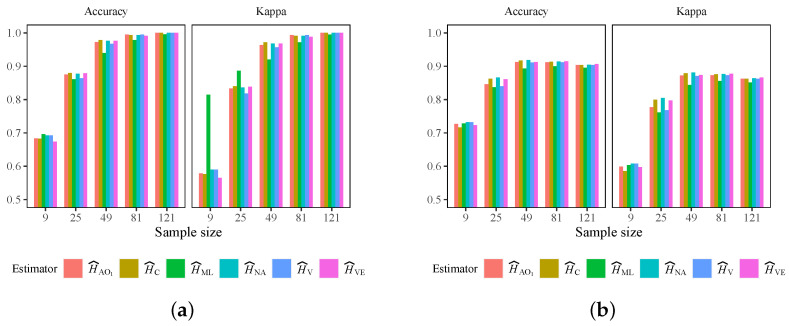
Accuracy and κ coefficient for SVM applied to simulated data with L=2. (**a**) Using testing set. (**b**) Using the whole image.

**Figure 6 entropy-24-00509-f006:**
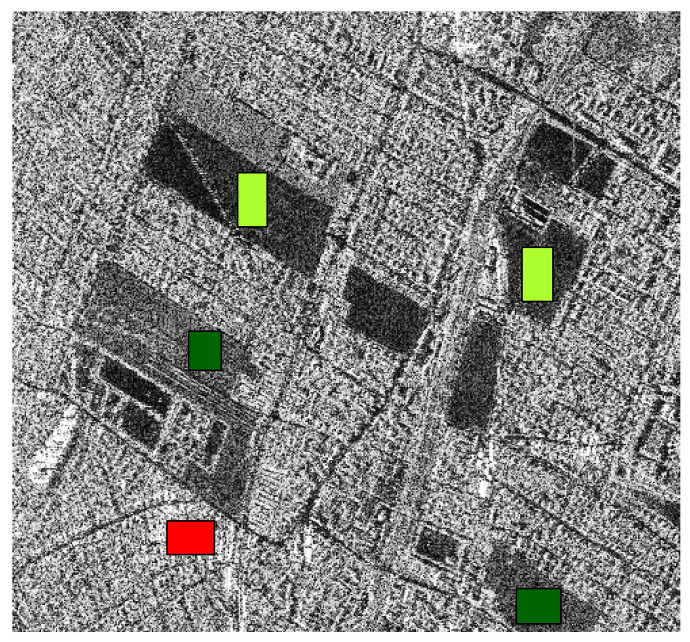
Image of the surrounding of Munich with reference samples.

**Figure 7 entropy-24-00509-f007:**
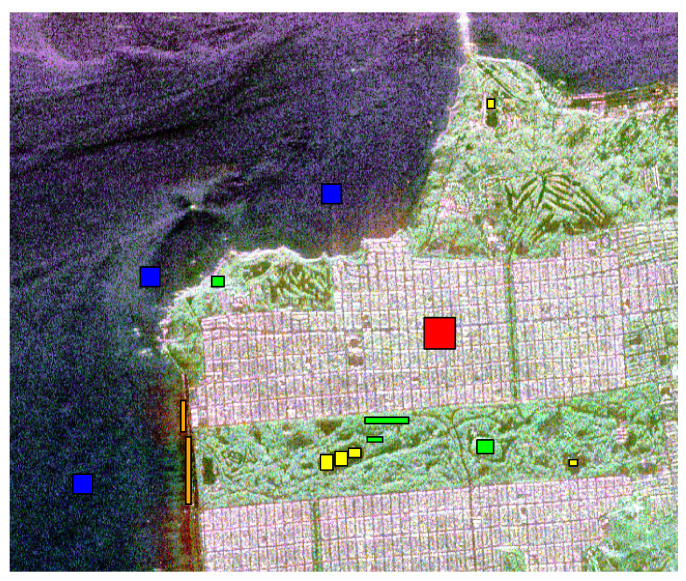
Image of San Francisco with reference samples.

**Figure 8 entropy-24-00509-f008:**
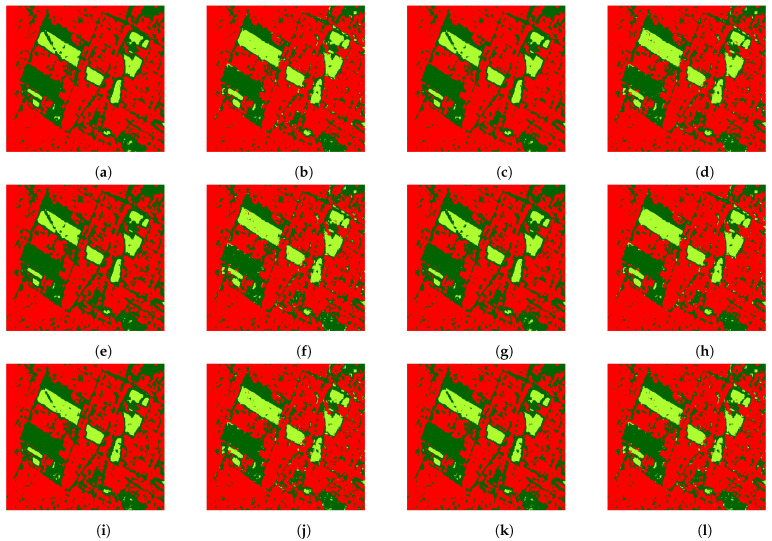
Surroundings of Munich classification using: (**a**) ${\hat{H}}_{{\mathrm{AO}}_{1}}$ (test acc. 0.9471), (**b**) ${\hat{H}}_{{\mathrm{AO}}_{1}}$ and CV (test acc. 0.9868), (**c**) ${\hat{H}}_{C}$ (test acc. 0.9394), (**d**) ${\hat{H}}_{C}$ and CV (test acc. 0.9923), (**e**) ${\hat{H}}_{\mathrm{NA}}$ (test acc. 0.9592), (**f**) ${\hat{H}}_{\mathrm{NA}}$ and CV (test acc. 0.9901), (**g**) ${\hat{H}}_{V}$ (test acc. 0.9526), (**h**) ${\hat{H}}_{V}$ and CV (test acc. 0.9846), (**i**) ${\hat{H}}_{\mathrm{VE}}$ (test acc. 0.9824), (**j**) ${\hat{H}}_{\mathrm{VE}}$ and CV (test acc. 0.9813), (**k**) ${\hat{H}}_{\mathrm{ML}}$ (test acc. 0.9713), and (**l**) ${\hat{H}}_{\mathrm{ML}}$ and CV (test acc. 0.9901).

**Figure 9 entropy-24-00509-f009:**
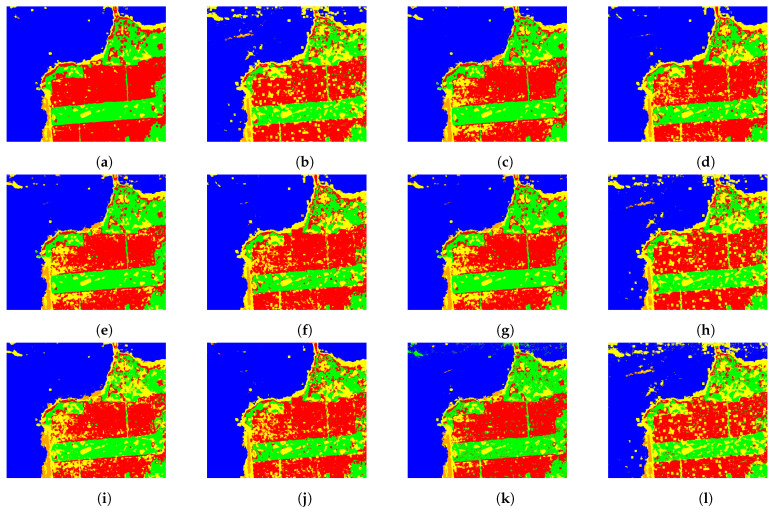
San Francisco classification using: (**a**) ${\hat{H}}_{{\mathrm{AO}}_{1}}$ (test acc. 0.9301), (**b**) ${\hat{H}}_{{\mathrm{AO}}_{1}}$ and CV (test acc. 0.9976), (**c**) ${\hat{H}}_{C}$ (test acc. 0.9692), (**d**) ${\hat{H}}_{C}$ and CV (test acc. 0.9882), (**e**) ${\hat{H}}_{\mathrm{NA}}$ (test acc. 0.9716), (**f**) ${\hat{H}}_{\mathrm{NA}}$ and CV (test acc. 0.9964), (**g**) ${\hat{H}}_{V}$ (test acc. 0.9727), (**h**) ${\hat{H}}_{V}$ and CV (test acc. 0.9953), (**i**) ${\hat{H}}_{\mathrm{VE}}$ (test acc. 0.9799), (**j**) ${\hat{H}}_{\mathrm{VE}}$ and CV (test acc. 0.9941), (**k**) ${\hat{H}}_{\mathrm{ML}}$ (test acc. 0.9585), and (**l**) ${\hat{H}}_{\mathrm{ML}}$ and CV (test acc. 0.9810).

**Figure 10 entropy-24-00509-f010:**
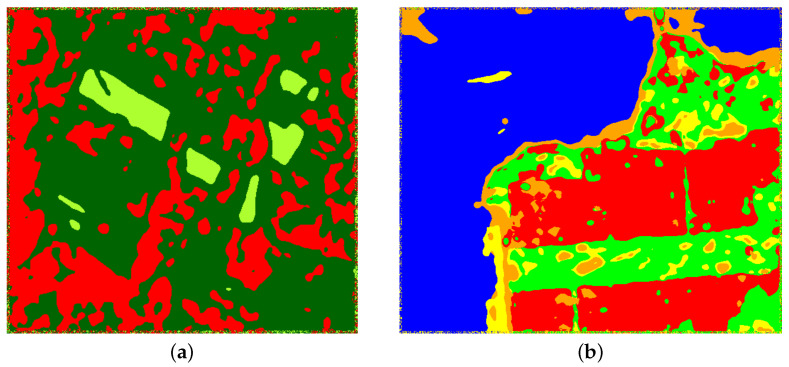
Classification using SVM after applying Lee filter to the image of: (**a**) Munich, (**b**) San Francisco.

**Table 1 entropy-24-00509-t001:** Selection criteria for the best *m* for each *n* and each entropy estimator, with $\alpha_{1}=-1.5$, $\alpha_{2}=-3$, $\alpha_{3}=-5$ and $\alpha_{4}=-8$.

*m*	$\alpha_{1}$	$\alpha_{2}$	$\alpha_{3}$	$\alpha_{4}$	$\bar{\hat{B}}$
1	${\hat{B}}_{11}$	${\hat{B}}_{12}$	${\hat{B}}_{13}$	${\hat{B}}_{14}$	${\bar{\hat{B}}}_{1.}=\frac{\sum_{j=1}^{4}\left|{\hat{B}}_{1j}\right|}{4}$
⋮	⋮	⋮	⋮	⋮	⋮
*s*	${\hat{B}}_{s1}$	${\hat{B}}_{s2}$	${\hat{B}}_{s3}$	${\hat{B}}_{s4}$	${\bar{\hat{B}}}_{s.}=\frac{\sum_{j=1}^{4}\left|{\hat{B}}_{ij}\right|}{4}$
⋮	⋮	⋮	⋮	⋮	⋮
⌊n/2⌋	${\hat{B}}_{\lfloor n/2\rfloor 1}$	${\hat{B}}_{\lfloor n/2\rfloor 2}$	${\hat{B}}_{\lfloor n/2\rfloor 3}$	${\hat{B}}_{\lfloor n/2\rfloor 4}$	${\bar{\hat{B}}}_{\lfloor n/2\rfloor .}=\frac{\sum_{j=1}^{4}\left|{\hat{B}}_{\lfloor n/2\rfloor j}\right|}{4}$
					$m=\underset{s}{arg min}{\bar{\hat{B}}}_{s.}$

**Table 2 entropy-24-00509-t002:** Heuristic spacing $m_{\mathrm{WG}}$, and best *m* chosen for each *n* and entropy estimator.

*L*	*n*	$m_{\mathrm{WG}}$	${\hat{H}}_{{\mathrm{AO}}_{1}}$	${\hat{H}}_{C}$	${\hat{H}}_{\mathrm{NA}}$	${\hat{H}}_{V}$	${\hat{H}}_{\mathrm{VE}}$
1	9	4	4	4	3	4	4
	25	6	6	5	3	8	2
	49	8	7	5	4	9	2
	81	10	7	4	5	8	2
	121	12	9	4	6	11	2
2	9	4	4	4	3	3	2
	25	6	8	4	4	9	2
	49	8	8	4	5	9	2
	81	10	9	4	5	9	2
	121	12	10	5	6	10	2

**Table 3 entropy-24-00509-t003:** Accuracy for *k*-means (k=4) applied to simulated data with L=2. Best values marked in bold.

*n*	${\hat{H}}_{{\mathrm{AO}}_{1}}$	${\hat{H}}_{C}$	${\hat{H}}_{\mathrm{NA}}$	${\hat{H}}_{V}$	${\hat{H}}_{\mathrm{VE}}$	${\hat{H}}_{\mathrm{ML}}$
9	0.664	0.659	0.665	0.665	0.640	0.682
25	0.839	0.855	0.859	0.837	0.846	0.796
49	0.911	0.915	0.915	0.909	0.911	0.884
81	0.916	**0.918**	**0.918**	0.916	**0.918**	0.900
121	0.905	0.906	0.907	0.905	0.907	0.895

**Table 4 entropy-24-00509-t004:** Classification quality indexes for *k*-means (k=4) applied to simulated data with n=81 and L=2. Best values marked in bold.

Index	${\hat{H}}_{{\mathrm{AO}}_{1}}$	${\hat{H}}_{C}$	${\hat{H}}_{\mathrm{NA}}$	${\hat{H}}_{V}$	${\hat{H}}_{\mathrm{VE}}$	${\hat{H}}_{\mathrm{ML}}$
CH	852,914	898,079	**902,719**	852,914	867,746	703,774
DB	0.441	0.434	**0.433**	0.441	0.442	0.467

**Table 5 entropy-24-00509-t005:** Best kernel (L: lineal, R: radial) and hyper-parameters for SVM applied to simulated SAR data.

*n*	${\hat{H}}_{{\mathrm{AO}}_{1}}$	${\hat{H}}_{C}$	${\hat{H}}_{\mathrm{NA}}$	${\hat{H}}_{V}$	${\hat{H}}_{\mathrm{VE}}$	${\hat{H}}_{\mathrm{ML}}$
9	R, c=5, g=1	R, c=1, g=0.1	R, c=5, g=2	R, c=5, g=2	L, c=0.1	R, c=10, g=0.01
25	R, c=5, g=2	L, c=10	L, c=10	R, c=10, g=1	R, c=1, g=1.5	L, c=0.1
49	L, c=10	L, c=10	R, c=10, g=1.5	R, c=5, g=1.5	L, c=0.1	R, c=5, g=1
81	R, c=10, g=2	L, c=5	L, c=10	R, c=10, g=2	R, c=1, g=2	L, c=1
121	L, c=5	L, c=5	L, c=1	L, c=1	L, c=1	L, c=0.01

**Table 6 entropy-24-00509-t006:** Best accuracy values and best models for SVM applied to simulated SAR data.

*n*	L=1	L=2
9	0.677, ${\hat{H}}_{{\mathrm{AO}}_{1}}$-${\hat{H}}_{V}$	0.732, ${\hat{H}}_{\mathrm{NA}}$-${\hat{H}}_{V}$
25	0.804, ${\hat{H}}_{\mathrm{NA}}$	0.866, ${\hat{H}}_{\mathrm{NA}}$
49	0.872, ${\hat{H}}_{C}$-${\hat{H}}_{\mathrm{NA}}$	0.918, ${\hat{H}}_{\mathrm{NA}}$
81	0.893, ${\hat{H}}_{V}$	0.915, ${\hat{H}}_{\mathrm{VE}}$
121	0.889, ${\hat{H}}_{C}$-${\hat{H}}_{V}$	0.907, ${\hat{H}}_{\mathrm{VE}}$

**Table 7 entropy-24-00509-t007:** Confusion matrices for synthetic data with L=1 (in percentage). Best values marked in bold.

		Reference			
	Prediction	α=−1.5	α=−3	α=−5	α=−8
${\hat{H}}_{{\mathrm{AO}}_{1}}$	α=−1.5	93.30	0.00	0.00	0.01
	α=−3	6.69	93.84	1.24	3.34
	α=−5	0.01	6.15	93.88	10.38
	α=−8	0.00	0.01	4.87	86.27
${\hat{H}}_{C}$	α=−1.5	**93.44**	0.01	0.00	0.01
	α=−3	6.55	93.31	1.12	3.04
	α=−5	0.01	6.67	93.82	10.38
	α=−8	0.00	0.01	5.06	86.57
${\hat{H}}_{\mathrm{NA}}$	α=−1.5	93.05	0.00	0.00	0.01
	α=−3	6.95	93.44	1.11	3.11
	α=−5	0.01	6.55	**94.50**	10.86
	α=−8	0.00	0.01	4.39	86.03
${\hat{H}}_{V}$	α=−1.5	93.29	0.00	0.00	0.01
	α=−3	6.71	**93.91**	1.30	3.45
	α=−5	0.01	6.07	93.84	10.44
	α=−8	0.00	0.01	4.86	86.09
${\hat{H}}_{\mathrm{VE}}$	α=−1.5	92.69	0.01	0.00	0.01
	α=−3	7.29	92.37	1.31	2.70
	α=−5	0.01	7.61	92.50	10.54
	α=−8	0.00	0.01	6.18	86.75
${\hat{H}}_{\mathrm{ML}}$	α=−1.5	92.99	0.00	0.00	0.00
	α=−3	7.00	93.63	0.87	2.84
	α=−5	0.01	6.36	94.47	10.34
	α=−8	0.00	0.01	4.66	**86.82**

**Table 8 entropy-24-00509-t008:** Confusion matrices for synthetic data with L=2 (in percentage). Best values marked in bold.

		Reference			
	Prediction	α=−1.5	α=−3	α=−5	α=−8
${\hat{H}}_{{\mathrm{AO}}_{1}}$	α=−1.5	**96.79**	0.17	0.00	0.09
	α=−3	3.20	93.40	0.44	3.54
	α=−5	0.01	6.43	96.76	9.21
	α=−8	0.00	0.00	2.80	87.16
${\hat{H}}_{C}$	α=−1.5	96.48	0.10	0.00	0.03
	α=−3	3.51	93.36	0.26	3.24
	α=−5	0.01	6.54	97.29	9.47
	α=−8	0.00	0.00	2.45	87.25
${\hat{H}}_{\mathrm{NA}}$	α=−1.5	96.48	0.12	0.00	0.03
	α=−3	3.51	93.36	0.25	3.25
	α=−5	0.01	6.52	97.39	9.48
	α=−8	0.00	0.00	2.35	87.23
${\hat{H}}_{V}$	α=−1.5	**96.79**	0.17	0.00	0.09
	α=−3	3.20	93.40	0.44	3.54
	α=−5	0.01	6.43	96.76	9.21
	α=−8	0.00	0.00	2.80	87.16
${\hat{H}}_{\mathrm{VE}}$	α=−1.5	96.08	0.10	0.00	0.02
	α=−3	3.91	93.64	0.27	3.00
	α=−5	0.01	6.26	97.07	9.35
	α=−8	0.00	0.00	2.66	87.63
${\hat{H}}_{\mathrm{ML}}$	α=−1.5	95.68	0.01	0.00	0.01
	α=−3	4.31	**93.96**	0.12	3.23
	α=−5	0.01	6.03	**97.42**	8.90
	α=−8	0.00	0.00	2.46	**87.86**

**Table 9 entropy-24-00509-t009:** Validation–test accuracy and Kappa coefficient values in the test set for the image of Munich. Best values marked in bold.

Feature Set	Model	Validation Accuracy	Test Accuracy	Kappa
Entropy estimator	${\hat{H}}_{{\mathrm{AO}}_{1}}$	0.9503	0.9471	0.9174
	${\hat{H}}_{C}$	0.9530	0.9394	0.9045
	${\hat{H}}_{\mathrm{NA}}$	0.9461	0.9592	0.9364
	${\hat{H}}_{V}$	0.9489	0.9526	0.9262
	${\hat{H}}_{\mathrm{VE}}$	0.9779	**0.9824**	**0.9722**
	${\hat{H}}_{\mathrm{ML}}$	0.9751	0.9713	0.9548
Entropy estimator and CV	${\hat{H}}_{{\mathrm{AO}}_{1}}$	0.9751	0.9868	0.9794
	${\hat{H}}_{C}$	0.9807	**0.9923**	**0.9878**
	${\hat{H}}_{\mathrm{NA}}$	0.9793	0.9901	0.9845
	${\hat{H}}_{V}$	0.9903	0.9846	0.9757
	${\hat{H}}_{\mathrm{VE}}$	0.9848	0.9813	0.9707
	${\hat{H}}_{\mathrm{ML}}$	0.9724	0.9901	0.9843

**Table 10 entropy-24-00509-t010:** Validation–test accuracy and Kappa coefficient values in the test set for the image of San Francisco. Best values marked in bold.

Feature Set	Model	Validation Accuracy	Test Accuracy	Kappa
Entropy estimator	${\hat{H}}_{{\mathrm{AO}}_{1}}$	0.9377	0.9301	0.9108
	${\hat{H}}_{C}$	0.9718	0.9692	0.9608
	${\hat{H}}_{\mathrm{NA}}$	0.9748	0.9716	0.9637
	${\hat{H}}_{V}$	0.9614	0.9727	0.9655
	${\hat{H}}_{\mathrm{VE}}$	0.9733	**0.9799**	**0.9743**
	${\hat{H}}_{\mathrm{ML}}$	0.9525	0.9585	0.9471
Entropy estimator and CV	${\hat{H}}_{{\mathrm{AO}}_{1}}$	0.9970	**0.9976**	**0.9970**
	${\hat{H}}_{C}$	0.9970	0.9882	0.9849
	${\hat{H}}_{\mathrm{NA}}$	0.9955	0.9964	0.9955
	${\hat{H}}_{V}$	1.0000	0.9953	0.9940
	${\hat{H}}_{\mathrm{VE}}$	0.9941	0.9941	0.9925
	${\hat{H}}_{\mathrm{ML}}$	0.9748	0.9810	0.9758

**Table 11 entropy-24-00509-t011:** Processing time to perform an entropy map with sliding windows of size 9×9. Best values marked in bold.

	Estimator					
**Image**	** ${\hat{H}}_{\mathrm{ML}}$ **	** ${\hat{H}}_{{\mathrm{AO}}_{1}}$ **	** ${\hat{H}}_{C}$ **	** ${\hat{H}}_{\mathrm{NA}}$ **	** ${\hat{H}}_{V}$ **	** ${\hat{H}}_{\mathrm{VE}}$ **
Munich	2.00	1.22	5.66	0.91	**0.85**	0.90
San Francisco	4.62	5.77	26.36	4.14	**4.03**	4.10

**Table 12 entropy-24-00509-t012:** Comparison results of our best proposal against an alternative method. Best values marked in bold.

Image	Model	Test Accuracy	Kappa
Munich	${\hat{H}}_{C}$ and CV	**0.9923**	**0.9878**
	Lee and SVM	0.9890	0.9829
San Francisco	${\hat{H}}_{{\mathrm{AO}}_{1}}$ and CV	**0.9976**	**0.9970**
	Lee and SVM	0.7606	0.6933

## Data Availability

The corresponding author will provide the data and code upon request.

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
