# Peer review of "Entropy Estimators in SAR Image Classification"

_entropy, 2022, doi:10.3390/e24040509_

Round 1
Reviewer 1 Report
The authors implemented an unsupervised (k-means) in synthetic images and supervised (SVM) classification algorithm in both simulated and actual data, and compared their performance with the entropy estimator. This paper is a little hard to understand; therefore, I have two suggestions that could be modified in this paper.
- Please give the number of each formula and cite them in this paper.
- Since this paper is difficult to understand, can the authors give some simple examples to illustrate the meaning of the formula or give examples if possible.
Reviewer 2 Report
In this paper, the effect of several Shannon entropy estimators on the performance of supervised and unsupervised SAR classification methods is studied. The relationship between the G0 and Fisher distributions is used to estimate the entropy for the parametric scenario. In the non-parametric scenario, bias, MSE, computational time, and accuracy are used for assessment. Empirical results on real-world data is provided to demonstrate that the method is effective. The paper is interesting and investigates a less-explored topic in the Remote Sensing community. I have the following comments, to be addressed before publication: 1. Comparison against alternative methods is limited. I would like 1-2 methods to be added for comparison and demonstrate that the proposed method offers advantages that prior methods cannot. 2. The introduction is too focused. There are works in the Remote Sending literature that try to address the challenges of processing SAR data using transfer learning methods. The following works needs to be discussed and compared against in the introduction: - Huang, Z., Pan, Z. and Lei, B., 2017. Transfer learning with deep convolutional neural network for SAR target classification with limited labeled data. Remote Sensing, 9(9), p.907. - Kang, C. and He, C., 2016, July. SAR image classification based on the multi-layer network and transfer learning of mid-level representations. In 2016 IEEE International Geoscience and Remote Sensing Symposium (IGARSS) (pp. 1146-1149). IEEE. - Rostami, M., Kolouri, S., Eaton, E. and Kim, K., 2019. Deep transfer learning for few-shot SAR image classification. Remote Sensing, 11(11), p.1374. - Lu, C. and Li, W., 2018. Ship classification in high-resolution SAR images via transfer learning with small training dataset. Sensors, 19(1), p.63. - Huang, Z., Dumitru, C.O., Pan, Z., Lei, B. and Datcu, M., 2020. Classification of large-scale high-resolution SAR images with deep transfer learning. IEEE Geoscience and Remote Sensing Letters, 18(1), pp.107-111. 3. In Figures 8 and 9, please also add a quantitative metric for comparison. A visual inspection might not be always feasible to determine the quality of algorithms. 4. On your tables, is it possible to add running time for algorithms. I think comparing in terms of computational complexity can be infromative.Author Response
Please see the attachment

Reviewer 3 Report
This paper provides a solid study to evaluate the performance of several parametric and non-parametric Shannon entropy estimators, by applying them as inputs to supervised and unsupervised classification algorithms. In addition, this paper proposes a methodology for fine-tuning non-parametric estimators of the entropy. Overall, the paper is well-written and justified by extensive experiments.
Some minor suggestions regarding the paper writing.
1. Figure 2: It would be good to provide reference numbers in the legend.
2. Please clearly list out the performance (e.g., bias, MSE) in a separate paragraph or a subsection.
Round 2
Reviewer 2 Report
The authors have addressed my concerns.